# A Low Cost Civil Vehicular Seamless Navigation Technology Based on Enhanced RISS/GPS between the Outdoors and an Underground Garage

**Ningbo Li [1], Lianwu Guan [1,*], Yanbin Gao [1,*], Zhejun Liu [1,2], Ye Wang [1,3] and Hanxiao Rong [1,4]**

[1] College of Automation, Harbin Engineering University, Harbin 150001, China; liningbo2016@hrbeu.edu.cn (N.L.); lzj1436791649@outlook.com (Z.L.); wangyeheu@hotmail.com (Y.W.); ronghanxiao@hotmail.com (H.R.)

[2] College of Computing, Queen's University, Kingston, ON K7L 3N6, Canada

[3] Department of Geomatics Engineering, University of Calgary, Calgary, AB T2N 1N4, Canada

[4] Department of Mechanical & Manufacturing Engineering, University of Calgary, Calgary, AB T2N 1N4, Canada

* Correspondence: guanlianwu@hrbeu.edu.cn (L.G.); yanbinbo9988@126.com (Y.G.); Tel.: +86-0451-82518042 (L.G. & Y.G.)

**Abstract:** Vehicles have to rely on satellite navigation in an open environment. However, satellite navigation cannot obtain accurate positioning information for vehicles in the interior of underground parking lots, as they comprise a semi-enclosed navigation space. Therefore, vehicular navigation needs to take into consideration both outdoor and indoor environments. Actually, outdoor navigation and indoor navigation require different positioning methods, and it is of great importance to choose a reasonable navigation and positioning algorithm solution for vehicles. Fortunately, the integrated navigation of the Global Positioning System (GPS) and the Micro-Electro-Mechanical System (MEMS) inertial navigation system could solve the problem of switching navigation algorithms in the entrance and exit of underground parking lots. This paper proposes a low cost vehicular seamless navigation technology based on the reduced inertial sensor system (RISS)/GPS between the outdoors and an underground garage. Specifically, the enhanced RISS is a positioning algorithm based on three inertial sensors and one odometer, which could achieve a similar location effect as the full model integrated navigation, reduce the costs greatly, and improve the efficiency of each sensor.

**Keywords:** vehicular seamless navigation technology; integrated navigation; enhanced reduced inertial sensor system (RISS)

## 1. Introduction

The development large cities leads to a sharp increase in the number of private cars, which has made the phenomenon of parking difficulties increasingly prominent. Meanwhile, due to the increase of the traffic pressure in cities, parking spaces are becoming scarce, coupled with the tension of China's urban land resources [1]. At present, car ownership in China has reached 217 million; the ratio of private cars to parking spaces in large cities is about 1:0.8, and that of small and medium cities is about 1:0.5. It is conservatively estimated that there is an insufficiency of more than 50 million car parking lots in China. Therefore, accelerating the construction of underground parking lots has also become a significant part of the plan in urban areas [2]. The outdoor positioning technologies, such as the Global Positioning System (GPS), can accurately provide location information for users in an open environment. However, due to the poor signal penetration capability of satellites, the satellite signals are blocked in indoor environments. Hence, indoor positioning data cannot be acquired correctly [3].

Actually, there are various types of indoor positioning techniques that can be used to compensate the error of the GPS data, such as visual positioning, laser navigation, mapping positioning, infrared ranging positioning, ultrasonic ranging positioning and inertial navigation positioning [4]. However, the low cost inertial navigation technology is most suitable for integration with satellite navigation due to their complementary characteristics in civil vehicular indoor and outdoor seamless navigation. Specifically, inertial sensors are mainly composed of gyroscopes and accelerometers. The navigation attitude information can be calculated by the internal sensors, and they need little environment data. However, when there are no calibration data, the positioning accuracy will diverge as the run time increases [5]. Therefore, the integration of satellite navigation and inertial navigation is the preferred method for connecting indoor and outdoor positioning [6]. When the system has the full inertial measurement unit (IMU) inertial navigation mode with a tri-axial gyroscope, tri-axial accelerometer and tri-axial magnetometer, the system can use the extended Kalman filter (EKF) for combined navigation with GPS satellite data and nine axis inertial sensor data [7]. However, due to some requirements such as those for civil vehicles, which should have a rigorous cost saving control for market competition, the traditional full inertial navigation system (INS)/GPS integrated navigation scheme will become less important. At this time, taking advantage of the enhanced reduced inertial sensor system (RISS) and GPS model for partial system integrated navigation, it is possible to achieve stable navigation with limited sensors [8].

The IMU with tri-axial accelerometers and tri-axial gyroscopes is the core component for an INS. Traditionally, the three-dimensional specific information is obtained by the three accelerometers. The three-dimensional rotation angular rates sensed by gyroscopes are adopted to obtain the attitude and transform the accelerometer measurements from the vehicle to the local level frame. Navigation information (attitude, velocity and position) is calculated by the outputs of gyroscopes and accelerometers, which is called the full IMU mechanization [9]. However, the full IMU with six inertial sensors is somewhat expensive for some low cost applications with medium precision requirements. What is more, the complete IMU with six inertial sensors is a highly redundant combination of sensors for a vehicle navigation system. To be specific, all six sensors do not need to be used at the same time in the actual vehicle navigation system [10]. Therefore, compared with traditional full IMU inertial sensors, some alternative solutions are explored, which can provide similar navigation information at a lower cost. This will make the micro-electro-mechanical system (MEMS) integrated sensor have less cost and enhance the efficiency for each sensor [11].

This paper explores and provides a full navigation solution only using one azimuth gyroscope, two horizontal accelerometers and one vehicle odometer. The enhanced RISS navigation algorithm can meet this requirement, which comprises the dead reckoning (DR) navigation theory [12]. The target of the solution is to achieve the scenario in which the land vehicles move mostly near the horizontal plan. In addition, the traditional EKF is a highly accurate and stable integrated navigation algorithm, which performs well on computer or high speed computing systems [4,13]. However, because of the complexity of the Kalman filter algorithm formula and the many related floating-point operations, the calculation will require a large amount of running time [14]. Therefore, this paper also explores a simple real-time algorithm to combine the GPS and enhanced RISS inertial navigation system. The improved algorithm can partially replace the attitude and positioning combination function of EKF and achieve similar effects. Under the premise of achieving similar integrated navigation effects, the updated navigation algorithm can further improve the operating efficiency of the integrated navigation algorithm and save a large amount of the cost. In addition, within the constraints of the costs and environment, the algorithm of this paper is only the initial algorithm of the navigation system. In the next stage, underground garage navigation can also introduce WiFi positioning, radio frequency identification (RFID) positioning, laser scanner simultaneous localization and mapping (SLAM) positioning and other sensor positioning, which can assist MEMS inertial navigation [15]. All the new sensors will further improve the positioning accuracy of underground garage navigation.

In Section 2, this paper establishes the navigation system structure and state update model. The state switching algorithm of the navigation system based on the enhanced RISS system is discussed in detail. The feasibility and the theory are verified with vehicle experiments in Section 3. This paper compares the environmental characteristics of the two navigation scenarios with two different filter navigation algorithms and shows the related experiment data and effect in each case. The result shows the positioning accuracy of the enhanced RISS simplified inertial navigation and the distinction between enhanced RISS and full IMU inertial navigation. The fourth section analyses the experimental results comprehensively, illustrating the reliability and robustness of the system from both the application and control perspectives.

## 2. Enhanced RISS Mechanization and Error Analysis

### 2.1. Enhanced RISS Navigation Mechanization

The enhanced RISS is comprised of one azimuth gyroscope providing the azimuth angular rate change, two accelerometers for calculating the pitch and roll angles in horizontal plan and an odometer with moving velocity in the near-horizontal plan [10]. First, the navigation system obtains the pitch and roll angle of the current vehicle platform through the angular relationship among the forward accelerometer, the transversal accelerometer and the local gravity acceleration. Then, the system takes advantage of the azimuth gyroscope and the initial heading angle to calculate the current azimuth angle of the vehicle. After the current attitude angle of the vehicular platform is obtained, the system uses the odometer on the vehicle to calculate the velocity of the current vehicle in various directions in the geographic frame system by conversion of the coordinate system. Because the system kinematic model is applicable to the East-North-Up (ENU) Earth fixed frame [16], the velocity in ENU directions will be required by the navigation algorithm. Finally, the latitude and longitude of the vehicle on the Earth can be required by combining the velocity integral with the feature constant of the Earth. The structure diagram of the enhanced RISS mechanization is shown in Figure 1 [16].

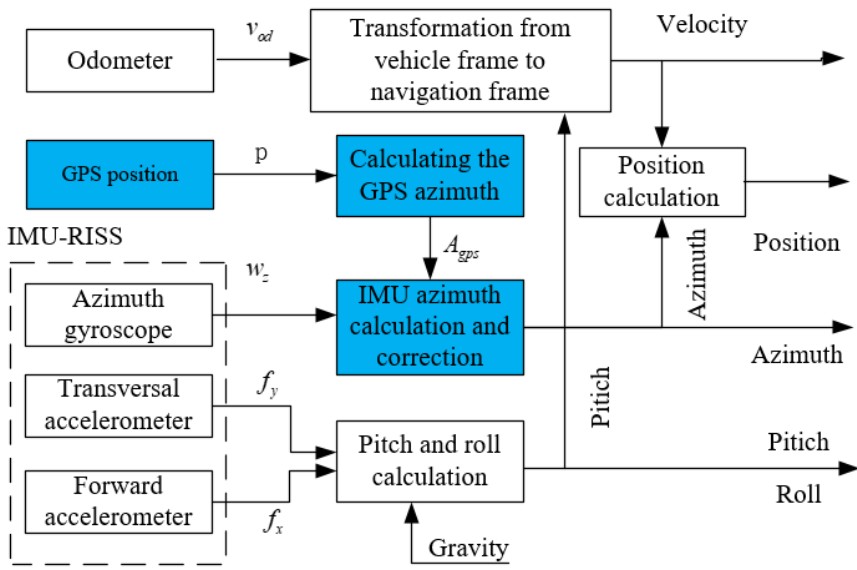

**Figure 1.** Structure diagram of the enhanced RISS mechanization.

During the enhanced RISS mechanization, the pitch of the vehicle is calculated by the forward accelerometer output information after error compensation. $f_x$ in the formula is the output of the system's forward accelerometer. $a_{od}$ is the velocity difference between the first and second epochs

of the vehicle odometer, which is also the current acceleration of the vehicle. Last, $g$ is the gravity acceleration of the current region, and the final pitch angle is calculated as [10]:

$$p = \sin^{-1}\left(\frac{f_x - a_{od}}{g}\right) \tag{1}$$

After that, the roll angle of the moving vehicle is calculated by the transversal accelerometer information $f_y$, the azimuth gyroscope measurement $w_z$ and the odometer velocity information. $v_{od}$ is the vehicle odometer output value, and $w_z$ is the measured value of the system azimuth gyroscope. Finally, the radian value of the roll angle is obtained by the inverse trigonometric function. Therefore, the roll angle of the navigation system calculation formula is [16]:

$$r = -\sin^{-1}\left(\frac{f_y + V_{od}w_z}{g\cos p}\right) \tag{2}$$

At the same time, the derivative of the system azimuth angle is mainly determined by the gyroscope output and the local latitude. $w_{ie}$ is the angular velocity of the Earth's rotation; $\varphi$ is the latitude of the Earth at the current location; and $V_e$ is the eastward velocity of $v_{od}$ after the decomposition. $R$ is the radius of the Earth. The formula of the derivative of the final azimuth angle is shown as [16]:

$$\dot{A} = -\left(w_z - w_{ie}\sin\varphi - \frac{V_e\tan\varphi}{R+h}\right) \tag{3}$$

After the attitude calculation of the navigation system, the velocity in local level frame $v_{od}$ can be derived with the following matrix:

$$V = \begin{bmatrix} V_e \\ V_n \\ V_u \end{bmatrix} = \begin{bmatrix} V_{od}\sin A\cos p \\ V_{od}\cos A\cos p \\ V_{od}\sin p \end{bmatrix} \tag{4}$$

Finally, the real position of the vehicle platform can be obtained by enhanced RISS integrated navigation. In Equations (5)–(7), $\dot{h}$ is the derivative of the height. $\dot{\varphi}$ is the derivative of the north movement, and $\dot{\lambda}$ is the derivative of the east movement. The step-by-step computation of the position is yielded by:

$$\dot{h} = V_u \tag{5}$$

$$\dot{\varphi} = \frac{V_n}{R+h} \tag{6}$$

$$\dot{\lambda} = \frac{V_e}{(R+h)\cos\varphi} \tag{7}$$

According to the above derivation, the attitude, velocity and latitude and longitude position of the vehicle can be obtained by the enhanced RISS navigation scheme. However, the initial latitude and longitude of the vehicle system can only be acquired by artificial operation [17]. If the system only relies on the enhanced RISS navigation positioning algorithm, after a period of time, the azimuth gyroscope integral of the inertial navigation system will diverge, according to the algorithm's derivation. Therefore, the system azimuth error will gradually accumulate, which will cause the positioning error to increase over time and large system errors to occur. Therefore, it is necessary to introduce the GPS satellite navigation and positioning system to correct and update the system velocity and position in the outdoor environment [6]. The traditional integrated navigation algorithm is the EKF [4]. In the EKF equation, the state quantity of the system is the attitude, velocity, position and sensors' error, and the velocity and position of the satellite navigation system are used for the observation quantity of the system to perform system combination navigation [13]. However, in the low cost civil vehicle seamless navigation system, the inexpensive core controller cannot provide the high

speed calculation and large capacity storage space for the algorithm. The EKF integrated navigation algorithm will reduce the low cost system's running efficiency, resulting in the slow transmission velocity of navigation data and untimely location update.

In this paper, according to the actual needs of the system, since the pitch and roll angles of the system are obtained from the accelerometers, the accelerometer sensor does not accumulate errors over time. Besides, the velocity of the vehicle is an accurate and stable measurement provided by the vehicle odometer. Therefore, the errors of the pitch angle, roll angle and the velocity of the vehicle do not accumulate. The cumulative error in the system occurs in the calculation of the azimuth angle, related to the azimuth gyroscope, which leads to the error of the decomposition of the velocity in the geographical coordinates. Finally, the cumulative error reflects the calculation of the latitude and longitude position of the system.

In order to decrease the cumulative error of positioning, this paper introduces the enhanced RISS algorithm. It is effective at correcting the error of the azimuth angle of the system with the GPS satellite positioning data within a certain period of time. The enhanced RISS navigation algorithm introduces the azimuth information of GPS on the basis of the traditional enhanced RISS navigation algorithm, so as to correct and supplement the divergent enhanced RISS azimuth angle. Meanwhile, it can simplify the related computation and enhance the efficiency of the sensors and controller, which avoids the complexity of the EKF integrated navigation algorithm. The target of the operation is to reduce the error accumulation of the system with time greatly. However, the satellite navigation system does not directly provide the azimuth information of the receiver. The system takes advantage of the Azimuth angle of GPS Point (AZP) algorithm to indicate the direction in which the vehicle travels [18]. The azimuth angle information is extracted from the GPS original positioning data, and it is defined as $A_{gps}$ calculated from the angle formed by the clockwise direction from the map of the north direction. After several experiments, when using a low cost GPS receiver chip, the system has only one GPS antenna. The AZP is a relatively simple and reliable algorithm to utilize the track to predict the azimuth angle of the vehicle [19]. However, the disadvantage of AZP is that the azimuth is inaccurate when the vehicle is started, but it was proven that the initial azimuth error does not affect the real-time running of the system. After the vehicle is driven, the azimuth error will be corrected swiftly. Leading the GPS azimuth data into the navigation system, the azimuth angle formula is calculated as:

$$A_{\text{t}} = A_{t-1} - \left( w_z - w_{ie} \sin \varphi - \frac{V_e \tan \varphi}{R + h} \right) \tag{8}$$

$$\theta_t = \arctan \frac{\text{Long}_t - \text{Long}_{t-1}}{\text{Lat}_t - \text{Lat}_{t-1}} \tag{9}$$

$$A_{gps} = \begin{cases} \theta_{t+1} - \theta_t & 0° \leq \theta_t < 360° \\ \theta_{t+1} - \theta_t + 360 & \theta_t < 0° \\ \theta_{t+1} - \theta_t - 360 & \theta_t \geq 360° \end{cases} \tag{10}$$

$$A = \gamma A_{gps} + (1 - \gamma) A_t \tag{11}$$

In Equations (9) and (10), $\theta_t$ is the azimuth angle of the vehicle calculated from the two adjacent GPS coordinate points when the GPS signal is in a good condition. It is the intersection angle between the current heading of the vehicle and the map of the north direction in the first time. $Long_t$ and $Lat_t$ represent the current coordinates of the longitude and latitude of the vehicle in a horizontal plane. The variables $t$ and $t-1$ are the two adjacent sampling points. $A_t$ is the azimuth angle from the enhanced RISS inertial navigation algorithm. Therefore, $A_{gps}$ is the azimuth angle from GPS positioning data. $\gamma$ is the weight value of the navigation system, which is used to regulate the weight of $A_{gps}$. After the real test in the vehicle platform, 0.8 is an adaptive value, which will reduce the influence of the cumulative error and promote the the precision of the positioning at the same time.

### 2.2. Garage Entrance Switching Decision

When the vehicle enters the garage, navigation accuracy will be reduced due to the GPS signal attenuation. Hence, to avoid this situation occurring, a switching algorithm, which can turn the integrated navigation mode into the DR navigation mode, should be applied at the underground garage entrance [20,21]. Moreover, we need some constraints to judge whether the navigation algorithm should be switched at one epoch or not. The current driving speed $V$ can be regarded as a key constraint through theoretical derivation and a series of experiments. When the vehicle decelerates and $V$ is less than half of the average speed $V_{av}$ and the initial speed $V_0$, the paper defines this state as a pre-switching state. $V_0$ will be equal to zero when the vehicle starts from a still position. At this moment, the vehicle heading is converted from the GPS track calculation to the DR algorithm in advance, while the way to obtain the position data does not change. When the system is running in the pre-switching state, the pitch angle $\theta_c$ of the vehicle and GPS parameters should be regarded as other constraints for the switching algorithm. Equation (12) is the pre-judgement of the current vehicle state. When $\theta_c$ is less than half of the underground garage inclination angle $\theta_g$ minus the equipment installation angle $\theta_f$, this means that the vehicle is going downhill in the garage. Nevertheless, we still need GPS parameters to make further judgement. When the position dilution of precision (PDOP) of the system reaches a low range and the satellite pulse per second (PPS) stops counting, the switching algorithm works, and the DR algorithm of the vehicle can run a stable and accurate trajectory in the underground garage [7]. This judgement process is reflected in Equation (13).

$$\begin{cases} V \leq \frac{V_0 + V_{av}}{2} \\ \\ 0 < \frac{V_n - V_{n+1}}{dT} \end{cases} \tag{12}$$

$$\begin{cases} \theta_c \leq \frac{\theta_g - \theta_f}{2} \\ PDOP < 7 \\ PPS = 0 \end{cases} \tag{13}$$

### 2.3. Errors' Analysis

#### 2.3.1. Attitude Errors' Analysis

The attitude errors of the system can be obtained by differentiating Equations (1)–(3) respectively and deducing the above formulas. The attitude errors of the system can be yielded as follows [22]:

$$\delta p = \frac{\delta f_x - \delta a_{od}}{g \cos p} - \frac{(f_x - a_{od}) \delta g}{g^2 \cos p} \tag{14}$$

$$\delta r = -\frac{\delta f_y + \delta V_{od} w_z + V_{od} \delta w_z}{g \cos p \cos r} + \frac{(f_y + V_{od} w_z)(\delta g \cos p - g \delta p \sin p)}{(g \cos p)^2 \cos r} \tag{15}$$

$$\delta \dot{A} = -\left( \delta w_z - w_{ie} \delta \varphi \cos \varphi - \frac{\delta V_e \tan \varphi + V_e \delta \varphi \sec^2 \varphi}{R + h} + \frac{V_e \tan \varphi \delta h}{(R + h)^2} \right) \tag{16}$$

From pitch error Equation (14), the pitch error $\delta p$ is determined by the forward accelerometer measurement error $\delta f_v$, the odometer measured forward accelerate error $\delta a_{od}$, and also, the Earth gravity error $\delta g$. Equation (15) shows the roll error. The roll error $\delta r$ is determined by the transversal accelerometer measurement error $\delta f_x$, odometer measurement error $\delta V_{od}$, vertical gyroscope measurement error $\delta w_z$, pitch error $\delta p$ and also, Earth gravity error $\delta g$. Equation (16) describes

the azimuth error. In the derivative of azimuth error $\delta\dot{A}$, taking into consideration that the Earth's radius $R$ is a relatively large value, the equation of the azimuth error can be simplified as:

$$\delta\dot{A} = \delta A_{gps} - (\delta w_z - w_{ie}\delta\varphi\cos\varphi) \tag{17}$$

Therefore, the azimuth error is mainly determined by the vertical gyroscope measurement error of the vehicle navigation system $\delta w_z$ and the latitude error $\delta\varphi$ in the geographic coordinate system.

### 2.3.2. Velocity Errors' Analysis

The velocity error equation can be required by differentiating Equation (4) and calculating the derivative of the error from the velocity formula. The velocity error can be yielded as follows:

$$\delta V_e = \delta V_{od}\sin A\cos p + \delta A V_{od}\cos A\cos p - \delta p V_{od}\sin A\sin p \tag{18}$$

$$\delta V_n = \delta V_{od}\cos A\cos p - \delta A V_{od}\sin A\cos p - \delta p V_{od}\cos A\sin p \tag{19}$$

$$\delta V_u = \delta V_{od}\sin p + \delta p V_{od}\cos p \tag{20}$$

Equations (18) and (19) are the eastward and northward velocity errors, respectively. From the equation, the eastward and northward velocity errors are mainly determined by the odometer measurement error $\delta V_{od}$, azimuth error $\delta A$ and pitch error $\delta p$ values. Equation (20) shows the upward velocity error, which is different from the eastward and northward direction. It is mainly determined by the odometer measurement error $\delta V_{od}$ and pitch error $\delta p$ values.

### 2.3.3. Position Errors' Analysis

Differentiating the height, latitude and longitude Equations (5)–(7), the derivative of position errors can be obtained as follows:

$$\delta\dot{h} = \delta V_u \tag{21}$$

$$\delta\dot{\varphi} = \frac{\delta V_n}{R+h} - \frac{V_n\delta h}{(R+h)^2} \tag{22}$$

$$\delta\dot{\lambda} = \frac{\delta V_e + \delta\varphi V_e\tan\varphi}{(R+h)\cos\varphi} - \frac{v_e\delta h}{(R+h)^2\cos\varphi} \tag{23}$$

From pitch error Equation (21), the derivative of the height error $\delta\dot{h}$ is only determined by the upward velocity error $\delta V_u$. Nevertheless, Equation (22) describes the derivative of the latitude error. The derivative of the latitude error $\delta\dot{\varphi}$ is mainly determined by the northward velocity error $\delta V_n$ and the height error $\delta h$. Lastly, Equation (23) represents the derivative of longitude error. The derivative of longitude $\delta\dot{\lambda}$ is mainly determined by the eastward velocity error $\delta V_e$ and the latitude error $\delta\varphi$.

## 3. Results and Error Analysis

This paper designs an underground navigation testing platform (UNTP) to test and analyse the accuracy and performance of our underground garage navigation method, named the simplified inertial navigation model RISS. The UNTP contains an MEMS (MPU9250, three axis gyroscope, three axis accelerometer inside, and three axis magnetometer), a vehicle bus decoding chip, a UBlox GPS chip (M8N series) and an ARM core MCU (STM32 ARM-M4). The enhanced RISS/GPS combination algorithm is processed in the micro-controller unit (MCU). We used the GPS single point positioning result and compensated the navigation by means of satellite navigation to compensate the azimuth angle. The host computer used the open source Windows map, which was used to display and record data from the interface during the experiment. First, we initialized the system in an open environment and made sure all the sensors were working stably. Then, it was essential to start the system, using the position and azimuth data obtained from the enhanced RISS/GPS integrated navigation system.

Lastly, the vehicle entered the underground garage. When the satellite signal was lost, the system updated the vehicle position and azimuth data according to the azimuth gyroscope and the odometer data. The initial position of the vehicle was mainly achieved from the GPS positioning point in the outdoor environment. If the vehicle were initialized in the underground garage, the initial point of the system would utilize the last coordinate point before the previous system termination.

This paper collects three groups of data, from the traditional Kalman Filter, enhanced RISS and the fibre inertial navigation system as the control group, and these three sets of experimental data were compared and the experimental conclusions drawn. These experiments were carried out around the 61st building of Harbin Engineering University and using the same actual vehicle based experimental platform to ensure stable and reliable experimental data. The UNTP structure of the experimental equipment is shown in Figure 2 [7]:

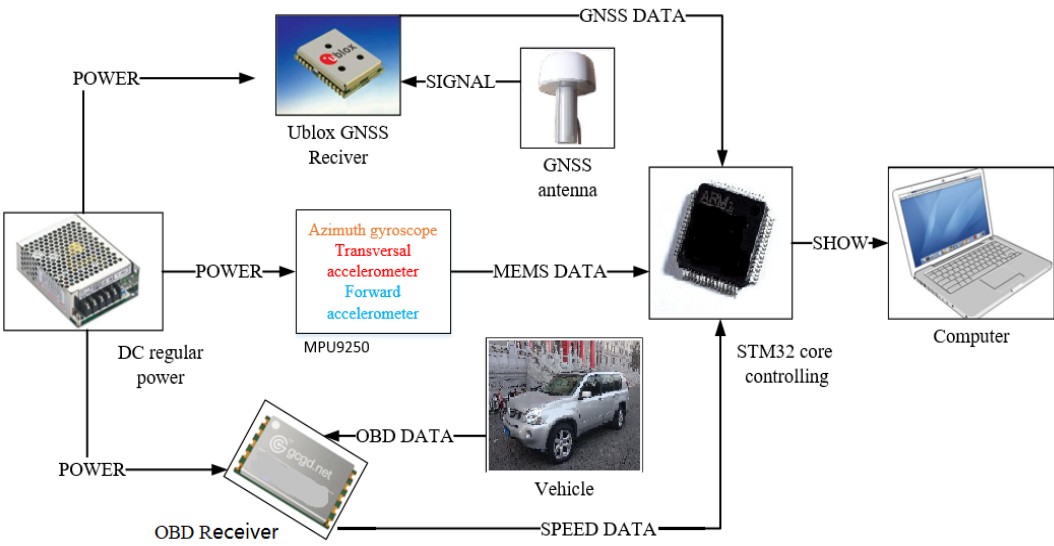

**Figure 2.** Experimental equipment structure diagram.

The vehicle experiment platform and the hardware of the navigation equipment used in the experiment are shown in Figure 3. The integrated navigation device was connected to the vehicle platform in a fixed manner. The signal antenna of GPS was installed on the top of the vehicle. The Y axis of the MEMS inertial sensors coincided with the direction of travel of the vehicle, which was convenient to calculate the azimuth angle. The upper software in the computer that was designed for this experiment was used to collect and process the navigation data. The vehicle speed was collected from the on board diagnostics (OBD) interface in the vehicle. The navigation system needed to decode the CAN bus protocol from the OBD interface to get the high accuracy vehicle speed, and the data came from the vehicle odometer. It could provide data support for the navigation algorithm in the underground garage.

Figure 4 is the waveform diagram of the pitch angle of the vehicle throughout the vehicle experimental platform. In the experimental diagram, the blue curve is the pitch angle of the EKF integrated navigation waveform, and the red curve is the pitch angle of the enhanced RISS inertial navigation pitch angle waveform. Comparing the two experimental waveforms, the experimental vehicle had an obvious negative angle around 60 s, indicating that the vehicle entered the downhill section of the underground garage. There was also a significant positive angle around 105 s, and the experimental vehicle went out of the underground garage. Comparing the noise of the two waveforms, it could be found that the noise of the extended Kalman filter was basically similar to the noise of enhanced RISS inertial navigation. At the entrance to the basement where the angle varied greatly, the noise of the enhanced RISS inertial navigation was slightly higher than that of the EKF

integrated navigation. This was mainly due to EKF integrated navigation relying on the full MEMS inertial navigation sensor. Taking advantage of the redundant two horizontal gyroscope compensation, the angle error effect was slightly better than the enhanced RISS inertial navigation angle calculation, using only uniaxial acceleration and odometer velocity. However, it could be seen from the effect of angle recognition that the speed of response of the two algorithm was the same, and the delay effect disappeared.

POWER  MEMS Navigation System   Fiber Navigation System                    Experiments platform

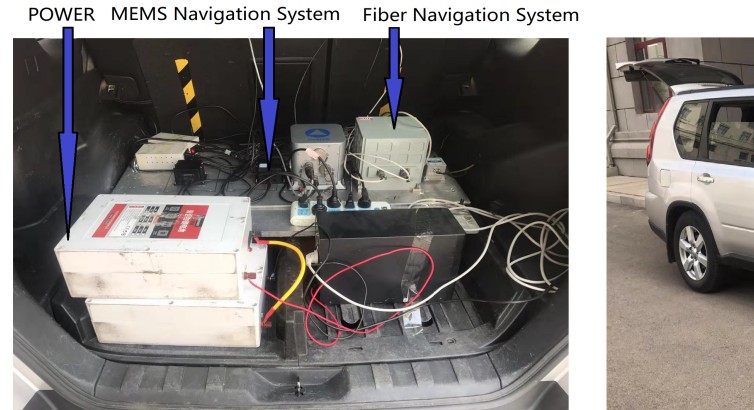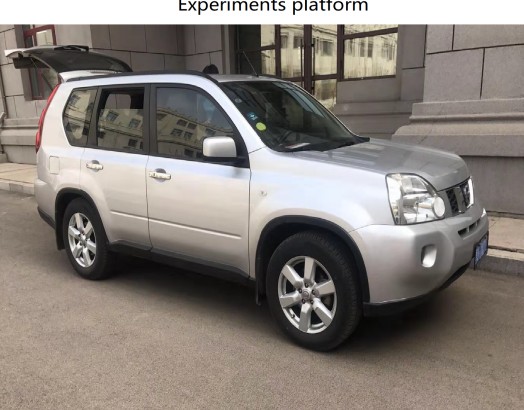

**Figure 3.** Experiments' platform.

The waveform diagram of the roll angle of the vehicle experimental platform is shown in Figure 5. It can be seen from the two waveforms that the fluctuation of the roll angle of the vehicle was relatively little during driving. Comparing the roll angle waveforms of the two algorithms, the noise of EKF was slightly smaller than the noise of enhanced RISS inertial navigation, which was similar to the pitch angle.

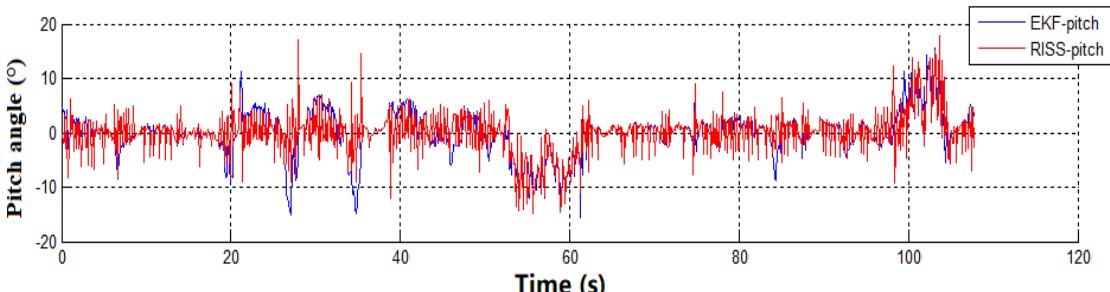

**Figure 4.** Extended Kalman filter (EKF)-Reduced Inertial Sensor System (RISS) pitch angle diagram.

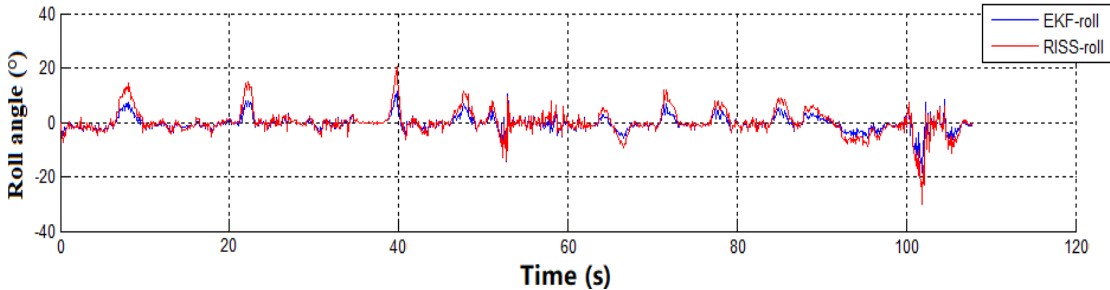

**Figure 5.** EKF-RISS roll angle diagram.

The waveform diagram of the azimuth angle of the vehicle experimental platform is shown in Figure 6. The blue curve in the figure is the azimuth angle waveform of the EKF integrated navigation,

and the red curve is the enhanced RISS navigation waveform. In Figure 6a, the enhanced RISS navigation system utilized only two horizontal accelerometers and azimuth gyroscope data to estimate the heading of the vehicle; while the GPS satellite navigation signal was introduced for heading correction in Figure 6b. It can be seen from Figure 6a that the vehicle was in the outdoor driving environment before 54 s, and the EKF navigation system used GPS positioning plus inertial navigation for integrated navigation. Compared with enhanced RISS navigation, the calculation of GPS azimuth needed to apply two GPS data points, and the updated frequency of the azimuth angle of GPS was 1 Hz. However, the updated frequency of the azimuth angle of the enhanced RISS was only related to the IIC communication frequency of the device, which was 400 kHz. Therefore, as shown in the waveform diagram in Figure 6a, around 60 s, the azimuth angle waveform of the EKF algorithm had a definite delay compared with the enhanced RISS azimuth angle waveform. However, the vehicle entered the underground parking lot after 54 s, and the GPS satellite signal disappeared. Both the EKF and RISS navigation algorithms only utilized the azimuth gyroscope to calculate the heading of the vehicle. This caused the delay effect of the azimuth angle to disappear. Since the enhanced RISS navigation algorithm calculated the heading angle of the vehicle by the azimuth gyroscope all the time, the bias of the azimuth angle appeared, which stemmed from the integral error accumulated over time. In contrast, because of the introduction of GPS data, the correction of the azimuth angle was performed with Equation (11). This proved that the enhanced RISS navigation algorithm could achieve a similar accuracy with the EKF integrated navigation algorithm, as shown in Figure 6b.

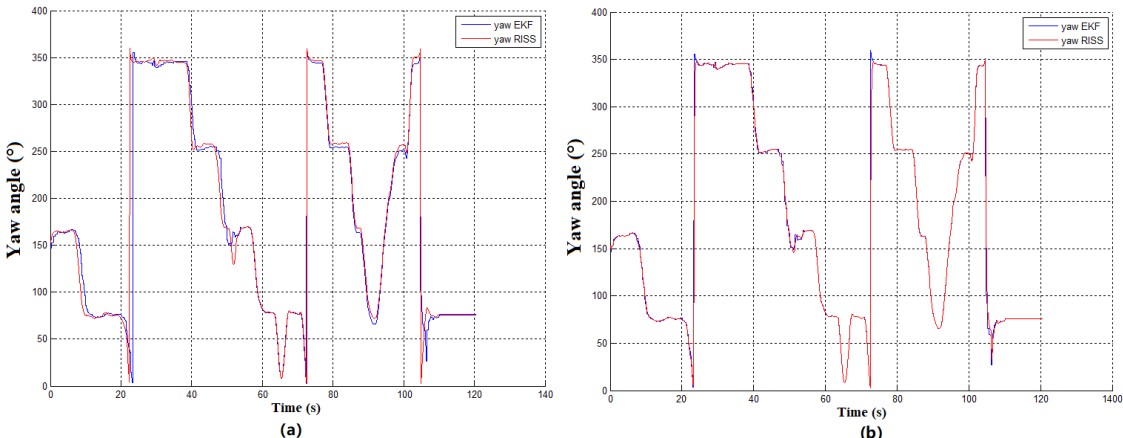

**Figure 6.** Yaw angle comparison diagram. (**a**) Yaw angle of RISS and EKF navigation system; (**b**) Yaw angle of enhanced RISS and EKF navigation system.

Figure 7 is the waveform diagram of the eastward velocity and the northward velocity in the ENU coordinate system. The blue curve in the figure is the velocity data acquired from the GPS position coordinates. It was obvious that the blue curve returned to zero after the oscillation for a few seconds from 54 s, which indicated that the GPS navigation data were stable to blurred and finally disappeared. The blue curve in the figure shows that the vehicle was in the process of entering the underground garage. After 105 s, the vehicle drove out of the underground parking lot, so that the GPS position and velocity data were reloaded in the outdoor environment. In the figure, the red curve is the east and north velocity waveforms, which were obtained from the EKF integrated navigation algorithm. The difference between the two figures is the green curve. The green curve in Figure 7 is the velocity data achieved from the traditional inertial sensor and the odometer. Due to the fact that there was no GPS satellite data to calibrate the azimuth angle, the azimuth gyroscope would accumulate bias over time. Finally, the accumulated bias led to the velocity data of the vehicle to diverge and have errors. In contrast, the RISS data in Figure 8 were corrected by the azimuth of the GPS, so that the drift of the azimuth angle was controlled. The component data of the decomposed odometer velocity eventually

converged to achieve the green curve velocity data with the enhanced RISS navigation algorithm, as shown in Figure 8.

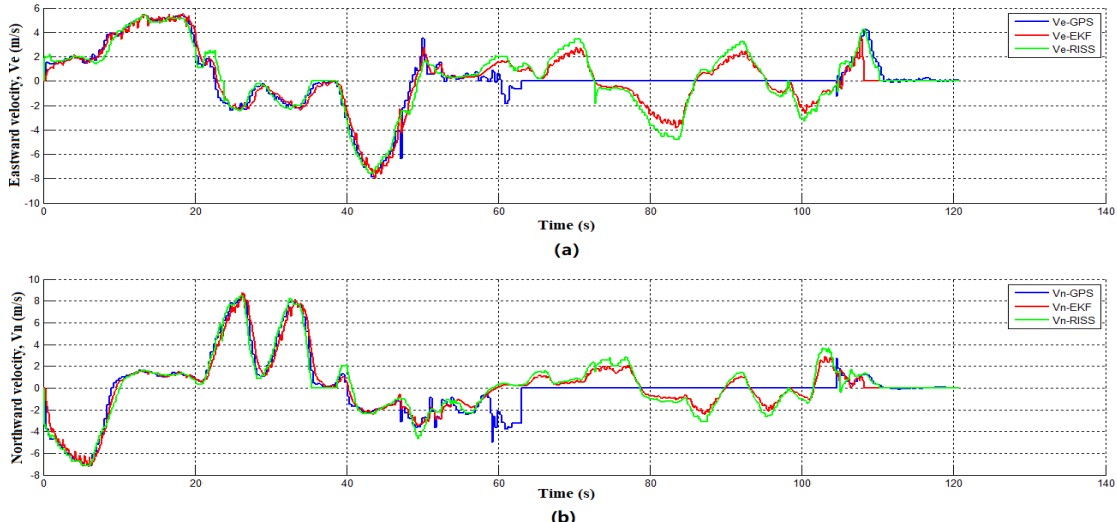

**Figure 7.** RISS velocity comparison diagram. (**a**) RISS eastward velocity comparison diagram; (**b**) RISS northward velocity comparison diagram.

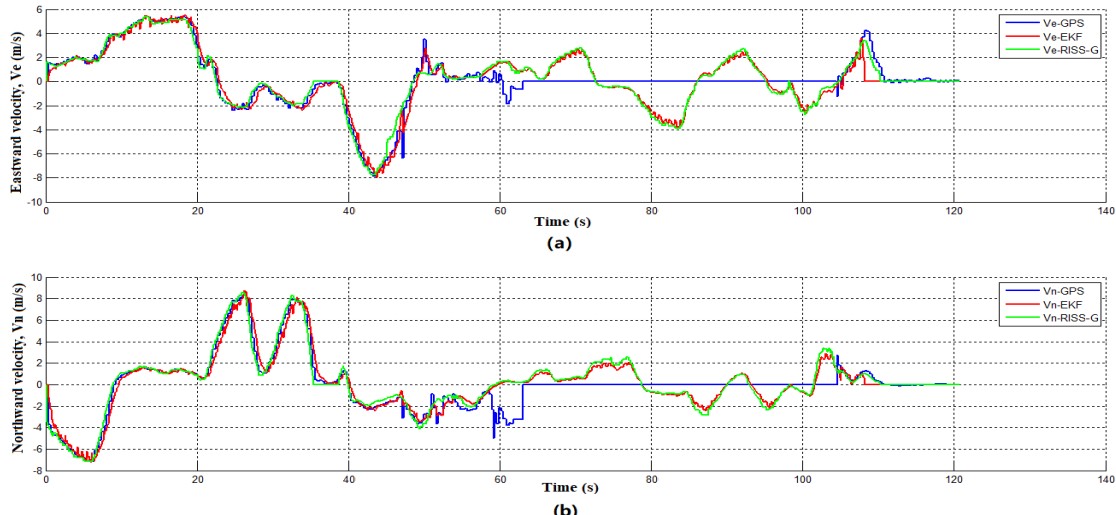

**Figure 8.** RISS/GPS velocity comparison diagram. (**a**) RISS/GPS eastward velocity comparison diagram; (**b**) RISS/GPS northward velocity comparison diagram.

In the navigation experiment, the fibre inertial navigation device was the reference system when it was compared with the MEMS navigation system. Figure 9 compares the trajectory of the fibre inertial navigation system with the trajectory of the enhanced RISS navigation algorithm at the same time. In Figure 9a, the red curve is the fibre inertial navigation track data, and the blue curve is the vehicle travel trajectory obtained by the RISS-DR algorithm. It can be shown from the figure that there was large bias in the trajectory of the vehicle of the RISS-DR algorithm, compared with the normal route. The error of the route could not be corrected and continued to expand. Finally, the position of the vehicle deviated more than 20 m. The cause of the bias of track was that the RISS-DR algorithm lacked the related correction sensor, such as GPS, and the heading gyroscope accumulated the error with time. The red curve in Figure 9b is consistent with Figure 9a, and the blue curve is the trajectory diagram of the enhanced RISS algorithm with the GPS integrated navigation. It can be seen from Figure 9b that, compared with the RISS-DR algorithm, the accuracy of the trajectory of

the RISS/GPS integrated navigation system was greatly improved. Therefore, the trajectory of the RISS/GPS integrated navigation system was close to the one of the fibre inertial navigation system. As a result, the enhanced RISS navigation algorithm with the GPS heading correction for integrated navigation could totally meet the demand of the positioning requirements of the underground garage navigation in the case of low cost navigation equipment.

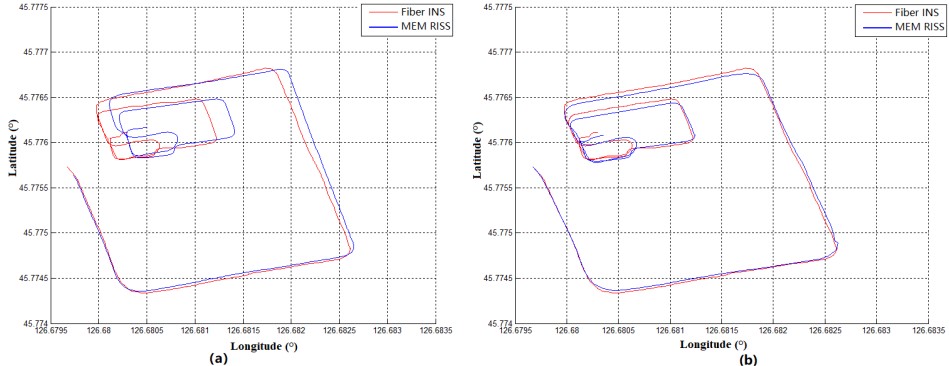

**Figure 9.** Fiber-INS MEMS-RISS trajectory comparison diagram. (**a**) Fiber-INS and RISS-DR trajectory diagram; (**b**) Fiber-INS and RISS-GPS trajectory diagram.

In this paper, the vehicular navigation system took advantage of the host computer software based on the Windows system, which was designed for the experiment to observe the real-time position of the vehicle navigation platform. The enhanced RISS algorithm and EKF algorithm data were collected in this experiment, as shown in Figure 10. The red curve in the system is the trajectory of the EKF filter algorithm from the vehicle, and the blue curve is GPS satellite positioning track data. It can be seen from the figure that the GPS trajectory of vehicle had a large scale positioning error when the vehicle entered the gate of the underground garage. After the vehicle entered the garage completely, the data disappeared. However, the integrated navigation positioning data were still working properly. When the vehicle left the underground garage, the positioning data of the two lines converged again. The vehicle navigation experiment proved the positioning stability of the integrated navigation system.

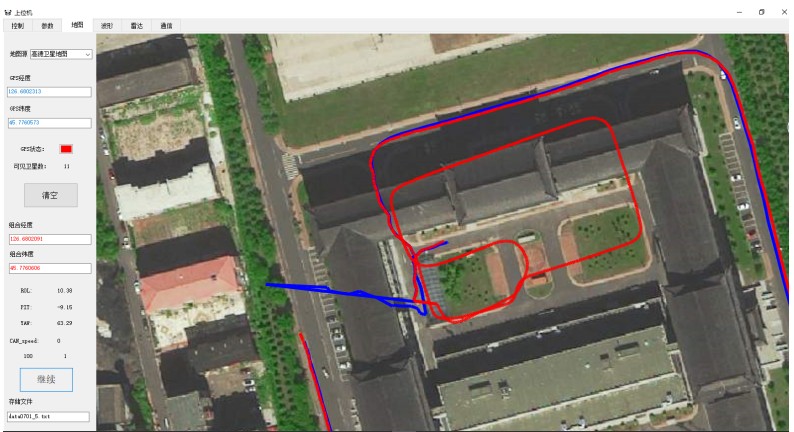

**Figure 10.** GPS-MEMS EKF trajectory comparison diagram.

According to the result of the comparison between the positioning data obtained by the enhanced MEMS RISS algorithm and the positioning data of the control group of the fibre inertial navigation, the position error of the RISS-DR algorithm and the RISS/GPS integrated navigation algorithm could be achieved, as shown in Figure 11a. Position error was the standard deviation of the position difference between RISS/GPS integrated navigation and fibre inertial navigation. It can be shown by the data in the figure that the positioning error of the two waveforms was less than 10 m in the

first 40 s. After 40 s, the RISS/GPS integrated navigation error kept stable basically, and the error of the RISS-DR algorithm accumulated with time and gradually increased. It showed a diverging trend. The large bias showed the necessity of integrated navigation for the enhanced RISS algorithm. Figure 11b is a control experiment of the yaw angle error between the RISS-DR algorithm and the RISS/GPS integrated navigation algorithm. The yaw angle error was the standard deviation of the yaw angle difference between RISS/GPS integrated navigation and fibre inertial navigation. It can be seen from the figure that the error of the yaw angle of the RISS-DR algorithm was always higher than the RISS/GPS integrated navigation yaw angle error and showed increasing trends in the future. The error of the RISS/GPS algorithm kept at a low level when the vehicle was moving, but the error increased at the beginning and end of the experiment, which indicated when the vehicle was static, the error of yaw increased. The detailed error parameters, about the position and yaw angle, are shown in Table 1. The phenomenon also indicated that the algorithm of the heading angle from GPS was more suitable for moving objects.

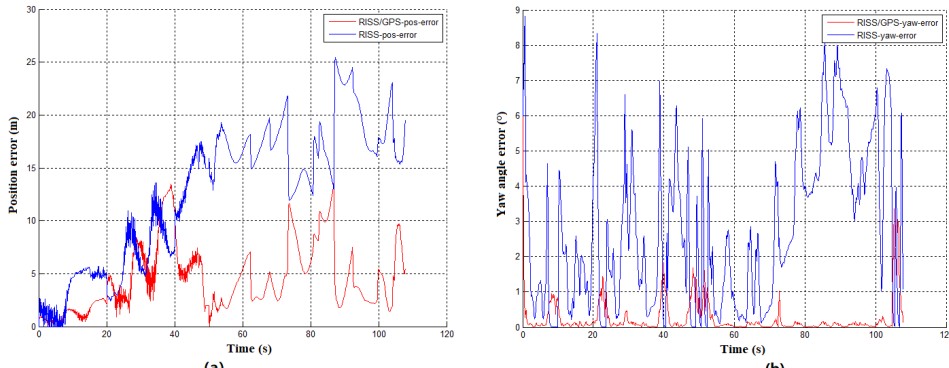

**Figure 11.** Position and yaw error diagram. (**a**) RISS-GPS and RISS-DR position error diagram; (**b**) RISS-GPS and RISS-DR yaw error diagram.

**Table 1.** The error parameter of the navigation system.

| Position Error Parameter | RISS-DR Algorithm | RISS-GPS Algorithm |
|---|---|---|
| Average | 12.6221 m | 4.0743 m |
| Maximum | 25.44569 m | 10.63819 m |
| Standard deviation | 6.5398 | 2.4085 |
| **Yaw angle error parameter** | **RISS-DR Algorithm** | **RISS-GPS Algorithm** |
| Average | 2.6914° | 0.2494° |
| Maximum | 8.83010° | 6° |
| Standard deviation | 2.18123 | 0.4705 |

The main accuracy parameters of the vehicular navigation system are shown in the Table 2.

**Table 2.** Navigation system experimental data accuracy.

| Accuracy Parameter | RISS-DR Algorithm | RISS-GPS Algorithm |
|---|---|---|
| Attitude | 5° | 5° |
| Velocity | 0.5 m/s | 0.1 m/s |
| Position | 20 m | 5 m |
| Yaw delay | 0 s | 0.5–0.8 s |

## 4. Conclusions

According to the structure of the enhanced RISS mechanization, the system could only utilize the measurements of two accelerometers, one gyroscope and one odometer to achieve navigation information output. Although the system accuracy of the system was acceptable for land vehicle navigation in tens of seconds, the RISS-DR algorithm would show a huge error in track location

due to the drift of the gyroscope, when the time is more than 60 s. However, after the GPS heading information compensation was combined with the enhanced RISS algorithm, the location accuracy of the experimental group of the RISS/GPS algorithm was greatly improved, compared with the control group of the fibre navigation system. In addition, plenty of complex algorithms are simplified compared to the EKF algorithm. The largest advantage of the enhanced RISS integrated navigation system was the low cost and the simplified algorithm, which was suitable for utilizing in miniaturized in-vehicle systems. The velocity accuracy was primarily determined based on the odometer output, which is often inaccurate because of the slipping of the wheels. Moreover, the drifts of the low precision inertial sensors used were the main error sources, which reduced the accuracy of the system to some extent.

RISS and GPS integrated navigation was an experimental and analytical test based on low cost considerations. The pitch angle and roll angle of the system achieved from the enhanced RISS algorithm could be compared with the EKF algorithm with the full IMU framework. The simplified system attitude error was increased to a certain extent compared with the full IMU system, but it showed little influence on the vehicle navigation system, which was not sensitive to the attitude. At the same time, the number of algorithms and sensor applications in the system was greatly simplified. Comparing the yaw angle data of the EKF and the enhanced RISS algorithm, it could be known that the integrated navigation enhanced RISS algorithm and the EKF algorithm had a slight delay, and the delay was in the range of 0.5–0.8 s, which mainly stemmed from the GPS transmission cycle. The delay of the GPS transmission cycle had little effect on the vehicle navigation system. Combined with the comprehensive judgement of the yaw angle error, after introducing the GPS heading angle, the system could greatly correct the yaw angle error that had diverged, so that the error of the yaw angle of the system was kept within two degrees in dynamic conditions. The compensation of GPS heading improved the positioning accuracy and stability of the navigation system obviously. It could be observed in the navigation system velocity waveform diagram that the GPS velocity lines fluctuated when the vehicle entered the gate of the underground garage. After the vehicle entered the garage completely, the GPS velocity curve became zero. At the same time, the EKF integrated navigation and enhanced RISS navigation ran normally. In addition, the RISS/GPS integrated navigation, compared with the RISS-DR velocity, kept the convergence, and the accuracy was higher than the RISS-DR velocity. Finally, through the comprehensive comparison of real-time data and analysis, the enhanced RISS navigation algorithm had many positive characteristics such as low dimensionality and quantity requirements for system sensors, low cost, and a simplified and efficient algorithm. However, only using the MEMS system for enhanced RISS algorithm could derive the position of the vehicle, which would make the location data of the system diverge. At this time, the introduction of GPS heading angle would assist with correcting the enhanced RISS yaw angle for integrated navigation. It could greatly improve the navigation accuracy, stability and robustness of the system without complications.

**Author Contributions:** N.L. and Y.G. conceived of the idea; N.L., Z.L. and Y.W. designed the software, collected the data and analysed the experimental data; L.G. collected the related resources and supervised the experiment; H.R. commented on the paper and proposed the experiment. All authors have read and agreed to the published version of the manuscript.

**Funding:** This research was supported by the National Natural Science Foundation of China (NSFC) (61803118), the Science and Technology Research Program of Chongqing Municipal Education Commission (KJZD-K201804701), the Post Doc. Foundation of Heilongjiang Province (LBH-Z17053) and the Fundamental Research Funds for the Central Universities (3072019CFM0402, 3072019CF0405).

**Conflicts of Interest:** The authors declare no conflict of interest.

**Abbreviations**

The following abbreviations are used in this manuscript:

RISS       Reduced Inertial Sensor System
IMU        Inertial Measurement Unit
INS        Inertial Navigation System
GPS        Global Positioning System
PDOP       Position Dilution Of Precision
PPS        Pulse Per Second
DR         Dead Reckoning
EKF        Extended Kalman Filter
MEMS       Micro-Electro-Mechanical System
AZP        Azimuth Angle of GPS Point
ENU        East-North-Up
OBD        On-Board Diagnostic
UNTP       Underground Navigation Testing Platform
MCU        Micro-Controller Unit

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
