# Peer review of "A Low Cost Civil Vehicular Seamless Navigation Technology Based on Enhanced RISS/GPS between the Outdoors and an Underground Garage"

_electronics, doi:10.3390/electronics9010120_

Round 1

Reviewer 1 Report

This paper aims to exploit the GPS azimuth to calibrate INS algorithm for enhancing positioning accuracy in the indoor environment. This idea is correct. The authors have completed the practical experiments to confirm this. In the following, there are some minor suggestions or questions for this paper before it is published.

1. Eq(9), what kinds of GPS observations you used for the θt computation? I think more detailed GPS equations are needed herein.

2.  There are different GPS positioning techniques e.g. SPP, PPP, RTK...,all of which can derive the azimuth angle.  Which techniques are used in this experiment? 

3. What kind of technique or method are employed to calculate the initial position of the vehicle?

4. In Section 2.3.3.,  only the derivatives of positions are described, why not the absolute positions? Figure 11 shows the derivate of 3D positions?

5. I am very interested in the degree of degradation of positioning accuracy in the indoor environment. Using the algorithm, how long the accuracy can be maintained within 10 m or larger?  

Author Response

Eq(9), what kinds of GPS observations you used for the θt computation? I think more detailed GPS equations are needed herein.

---Response: Thanks for your valuable attention about this issues. θt is the azimuth angle of the vehicle that calculated from the two adjacent GPS coordinate points when GPS signal is in good condition, which is yield by:

where, (, ) represent the current coordinate of Longitude and Latitude of the vehicle in a horizontal plane. t and t-1 are the two adjacent sampling point.

What worth note is that the azimuth angle of the vehicle calculated by the GPS coordinate points in the stationary state is not accurate, as stated in the article: “the disadvantage of AZP is that the azimuth is inaccurate when the vehicle is started, but it is proved that the initial azimuth error does not affect the real-time running of the vehicular navigation system”. The updated and corrected sentence is shown in the paper. Thanks for your advice.

There are different GPS positioning techniques e.g. SPP, PPP, RTK...,all of which can derive the azimuth angle. Which techniques are used in this experiment?

---Response: Thanks for your valuable comment. According to the experimental goals, all the devices in this experiment use low-cost commercial devices. This experiment uses the UBX communication protocol obtained by the U-Blox GPS receiver to achieve the GPS positioning points. The RISS algorithm only uses single point positioning (SPP) based simple GPS positioning. Therefore, we do not employ the RTK and PPP in the low-cost navigation application. Moreover, the azimuth angle precision that calculated by RTK or PPP would be much higher and the overall cost of the system would also be increased a lot according to the different precision requirements.

What kind of technique or method are employed to calculate the initial position of the vehicle?

--- Response: Thanks for your valuable comment. The initial coordinate position of the vehicle is directly achieved from the GPS positioning point in outdoor environment. Moreover, if the vehicle is initialized in the underground garage, the initial coordinate point of the system will utilize the last stored point before the previous system termination. The updated and corrected sentence is shown in the paper. Thanks for your advice.

In Section 2.3.3., only the derivatives of positions are described, why not the absolute positions? Figure 11 shows the derivate of 3D positions?

---Response: Thanks for your valuable comment. In this paper, the final position output of the integrated navigation system stems from the integral of the position derivative. So, the error equation that using the derivative forms, would make the error expression more accurate and clear. Figure 11 describes the Longitude and Latitude errors of the system achieved from the real-time calculations. When the vehicle enters the underground garage, the GPS signal of the vehicle is lost completely. After entering the underground garage, the system can only get two-dimensional position data with a single gyroscope and vehicle velocity. The height information actually cannot be received. In addition, the vehicle's outdoor altitude position has changed little and its state keeps stable. Therefore, the error waveform only shows the Latitude and Longitude position information that fluctuates significantly.

I am very interested in the degree of degradation of positioning accuracy in the indoor environment. Using the algorithm, how long the accuracy can be maintained within 10 m or larger?

--- Response: Thanks for your concerning. According to the experimental results, the vehicle runs 1 minute in the underground garage with the positioning error of 5m in indoor environment. The long-term cumulative error is similar with the result, which is about 5.2m per minute.

Reviewer 2 Report

This is an interesting work based on sensor (and GPS) fusion for vehicles navigation. The work is not original (low novelty) on its roots but presents interesting results avoiding the use of the well-known EKF. Nevertheless, the use EKF in vehicles will not be a problem in most of the current vehicles due to the fact that most of them already was a considerable amount of processing power onboard (and I’m not considering the ones with autonomous driving systems)… Please provide the figure numbers (e.g.: computation time) in terms of computational complexity for your algorithm and for the EKF.

Author Response

This is an interesting work based on sensor (and GPS) fusion for vehicles navigation. The work is not original (low novelty) on its roots but presents interesting results avoiding the use of the well-known EKF. Nevertheless, the use EKF in vehicles will not be a problem in most of the current vehicles due to the fact that most of them already was a considerable amount of processing power onboard (and I’m not considering the ones with autonomous driving systems)… Please provide the figure numbers (e.g.: computation time) in terms of computational complexity for your algorithm and for the EKF.

--- Response: Thanks for your valuable comment. According to the EKF and RISS algorithm used in this experiment, we tested the time-consumption of the two algorithms at running time aspect. The calculation time in offline MATLAB is 0.126s and 0.059s by using EKF and RISS, respectively. These results are recalculated with MATLAB from the actual data and algorithms, but a microcontroller based on ARM-M7 is used for online debugging and running. The time consuming may be different from the results actually. However, the comparison results show that RISS occupies less resources than EKF algorithm and takes less time in the low-cost devices and scenarios. In totally, RISS and EKF can achieve similar effort and RISS runs faster than EKF. Thanks for your advice.

Reviewer 3 Report

In my opinion, the reviewed paper is very interesting, however, its current version is should be improved before publication in the Electronics. Below, I present my suggestions for paper improvement:

1) Template - please see the Electronics template and use all rules to your paper, e.g., write the section and subsection captions from upper case letters except for preposition, articles, and conjunctions, etc. For example, the wrong formatted paragraphs are visible due to the lack of line numbering.

2) Brackets - give space before any brackets "(" or "[".

3) Introduction - At the final part of the introduction, in the paper layout description, the authors should use "Section 2", "Section 3", etc. instead of "the second section", "the third section", etc.

4) Figures especially with graphs should be improved:

a) In Figure 1, the authors use perhaps mistakenly twice "pitich" instead of "pitch".

b) In Figures 5-9 and 11 should be improved axis captions, i.e., each axis should describe by variable name, appropriate symbol (if possible), and unit, e.g., in Figure 4 should be y-axis: "Pitch angle (°)", x-axis: "Time (s)", in Figure 5 - y-axis: "Roll angle (°)", x-axis: "Time (s)", in Figure 6 - y-axis: "Yaw angle (°)", x-axis: "Time (s)", in Figure 7,8 (1) - y-axis: "Eastward velocity, Ve (m/s)", (2) - y-axis: "Northward velocity, Vn (m/s)", (1,2) x-axis: "Time (s)", in Figure 9 - y-axis: "Latitude (°)", x-axis: "Longitude (°)", in Figure 11 (1) - y-axis: "Position error (m)", (2) - y-axis: "Yaw angle error (°)", x-axis: "Time (s)".

c) In figures with two separate graphs, the authors should use the notation (a) and (b) instead of (1) and (2) (see Figures 6 and 9) - the same idea should be used in Figures 7 and 8. It is good practice to describe these variants in the figure caption.

5) In references to figures, tables, and equations the authors should not use definite articles "the", i.e., "in Figure 2" instead of "in the Figure 2" (see line 167 p. 7); "in Table 1" instead of "in the Table 1" (see line 273, p. 12); "differentiating Equations (1)-(3)" instead of "differentiating the equations (1),(2) and (3)" (see first sentence in 2.3.1, p. 6).

Please check the whole paper and improve these mistakes.

6) Equations and symbols:

a) All entered symbols should be explained, e.g., immediately after its presentation in equations.

b) Below Equation (1), the authors write "... transversal accelerometer information fx, ..." - there should be maybe "... transversal accelerometer information fy, ..." or "... forward accelerometer information fx, ...".

c) In Equations (3) and (8), there is the symbol "wie" but in the text above the Equation (3) is "wie".

d) In Equation (3), the authors use unexplained symbols "RN" and "h" and in Equation (6) and (7) - "R" - these symbols should be explained. RN=R? R is explained only above Equation (14).

e) Sometimes, the authors use a dot above the symbol, e.g., in Equations (5)-(6) - is a derivative over time? if yes, this should be explained.

f) In the conditions of Equation (9), there is an unexplained symbol "t" - maybe, there should be "θt"?

g) The following description of the formulas is unacceptable, e.g., "Equation (12) is the roll error equation." Do not use twice "equation" in the same sentence. For example, this sentence could be "Equation (12) describes/represents the roll error." etc. Please change the other similar sentences.

h) (p. 6, line 136) there should be "Equations (15) and (16) are the eastward and northward velocity errors, respectively." instead of "Equation (15) and Equation (16) is the Eastward velocity error and Northward velocity error." Please check the other sentence...

7) References: please improve the descriptions of the references, e.g., in [1] - the lack of pages. Include the digital object identifier (DOI) for all references where available. (see template).

7) Other suggestions:

a) PDOP abbreviation is not explained (see p. 6).

b) RISS abbreviation in Keywords should be explained.

8) English in the paper should be improved! You could use the free Grammarly software, see: http://www.grammarly.com.

Author Response

In my opinion, the reviewed paper is very interesting, however, its current version is should be improved before publication in the Electronics. Below, I present my suggestions for paper improvement:

Thank you very much for your insightful comments, which are of great value for us to improve this manuscript. According to your suggestions, some shortcomings are revealed in our current version of research work and we will improve our research level and achieve more fruitful results in future work. All of your suggestions have great significance for guiding our research work.

1) Template - please see the Electronics template and use all rules to your paper, e.g., write the section and subsection captions from upper case letters except for preposition, articles, and conjunctions, etc. For example, the wrong formatted paragraphs are visible due to the lack of line numbering.

---Response: Thanks for your suggestion, they are all corrected in the latest version of the paper.

2) Brackets - give space before any brackets "(" or "[".

---Response: Thanks for your suggestion, they are corrected in the latest version of the paper.

3) Introduction - At the final part of the introduction, in the paper layout description, the authors should use "Section 2", "Section 3", etc. instead of "the second section", "the third section", etc.

---Response: Thanks for your suggestion, they are corrected in the latest version of the paper.

4) Figures especially with graphs should be improved:

a) In Figure 1, the authors use perhaps mistakenly twice "pitich" instead of "pitch".

---Response: Thanks for your suggestion, they are corrected in the latest version of the paper. Moreover, we also checked the overall manuscript to avoid the mistakes like this.

b) In Figures 5-9 and 11 should be improved axis captions, i.e., each axis should describe by variable name, appropriate symbol (if possible), and unit, e.g., in Figure 4 should be y-axis: "Pitch angle (°)", x-axis: "Time (s)", in Figure 5 - y-axis: "Roll angle (°)", x-axis: "Time (s)", in Figure 6 - y-axis: "Yaw angle (°)", x-axis: "Time (s)", in Figure 7,8 (1) - y-axis: "Eastward velocity, Ve (m/s)", (2) - y-axis: "Northward velocity, Vn (m/s)", (1,2) x-axis: "Time (s)", in Figure 9 - y-axis: "Latitude (°)", x-axis: "Longitude (°)", in Figure 11 (1) - y-axis: "Position error (m)", (2) - y-axis: "Yaw angle error (°)", x-axis: "Time (s)".

---Response: Thanks for your suggestion, they are corrected in the latest version of the paper.

c) In figures with two separate graphs, the authors should use the notation (a) and (b) instead of (1) and (2) (see Figures 6 and 9) - the same idea should be used in Figures 7 and 8. It is good practice to describe these variants in the figure caption.

---Response: Thanks for your suggestion, they are corrected in the latest version of the paper.

5) In references to figures, tables, and equations the authors should not use definite articles "the", i.e., "in Figure 2" instead of "in the Figure 2" (see line 167 p. 7); "in Table 1" instead of "in the Table 1" (see line 273, p. 12); "differentiating Equations (1)-(3)" instead of "differentiating the equations (1),(2) and (3)" (see first sentence in 2.3.1, p. 6).

---Response: Thanks for your suggestion, they are corrected in the latest version of the paper.

Please check the whole paper and improve these mistakes.

6) Equations and symbols:

a) All entered symbols should be explained, e.g., immediately after its presentation in equations.

---Response: Thanks for your suggestion, they are corrected in the latest version of the paper.

b) Below Equation (1), the authors write "... transversal accelerometer information fx, ..." - there should be maybe "... transversal accelerometer information fy, ..." or "... forward accelerometer information fx, ...".

---Response: Thanks for your suggestion, they are corrected in the latest version of the paper.

c) In Equations (3) and (8), there is the symbol "wie" but in the text above the Equation (3) is "wie".

---Response: Thanks for your suggestion, they are corrected in the latest version of the paper.

d) In Equation (3), the authors use unexplained symbols "RN" and "h" and in Equation (6) and (7) - "R" - these symbols should be explained. RN=R? R is explained only above Equation (14).

---Response: Thanks for your suggestion, they are corrected in the latest version of the paper.

e) Sometimes, the authors use a dot above the symbol, e.g., in Equations (5)-(6) - is a derivative over time? if yes, this should be explained.

---Response: Thanks for your suggestion, they are corrected in the latest version of the paper.

f) In the conditions of Equation (9), there is an unexplained symbol "t" - maybe, there should be "θt"?

---Response: Thanks for your suggestion, they are corrected in the latest version of the paper.

g) The following description of the formulas is unacceptable, e.g., "Equation (12) is the roll error equation." Do not use twice "equation" in the same sentence. For example, this sentence could be "Equation (12) describes/represents the roll error." etc. Please change the other similar sentences.

---Response: Thanks for your suggestion, they are corrected in the latest version of the paper.

h) (p. 6, line 136) there should be "Equations (15) and (16) are the eastward and northward velocity errors, respectively." instead of "Equation (15) and Equation (16) is the Eastward velocity error and Northward velocity error." Please check the other sentence...

---Response: Thanks for your suggestion, they are corrected in the latest version of the paper.

7) References: please improve the descriptions of the references, e.g., in [1] - the lack of pages. Include the digital object identifier (DOI) for all references where available. (see template).

---Response: Thanks for your suggestion, they are corrected. The pages are updated in the manuscript, but the DOI cannot be achieved in Gooogle.

7) Other suggestions:

a) PDOP abbreviation is not explained (see p. 6).

---Response: Thanks for your suggestion, they are corrected in the latest version of the paper.

b) RISS abbreviation in Keywords should be explained.

---Response: Thanks for your suggestion, they are corrected in the latest version of the paper.

8) English in the paper should be improved! You could use the free Grammarly software, see: http://www.grammarly.com.

Round 2

Reviewer 2 Report

I accept the authors response to my comments. Nevertheless, I keep my opinion in relation to the presenting work that it is interesting work that avoids the use of the well-known EKF. However, the use EKF in vehicles is not a problem in most of the current vehicles due to the fact that most of them already was a considerable amount of processing power onboard (and I’m not considering the ones with autonomous driving systems)…